# Synthesis, structural analysis, and properties of highly twisted alkenes 13,13'-bis(dibenzo[*a,i*]fluorenylidene) and its derivatives

Hao-Wen Kang[1], Yu-Chiao Liu[2], Wei-Kai Shao[1], Yu-Chen Wei[3], Chi-Tien Hsieh[1], Bo-Han Chen[4], Chih-Hsuan Lu[4], Shang-Da Yang[4], Mu-Jeng Cheng [1], Pi-Tai Chou [3] ✉, Ming-Hsi Chiang [2,5] ✉ & Yao-Ting Wu [1] ✉

The rotation of a C = C bond in an alkene can be efficiently accelerated by creating the high-strain ground state and stabilizing the transition state of the process. Herein, the synthesis, structures, and properties of several highly twisted alkenes are comprehensively explored. A facile and practical synthetic approach to target molecules is developed. The twist angles and lengths of the central C = C bonds in these molecules are 36–58° and 1.40–1.43 Å, respectively, and confirmed by X-ray crystallography and DFT calculations. A quasi-planar molecular half with the π-extended substituents delivers a shallow rotational barrier (down to 2.35 kcal/mol), indicating that the rotation of the C = C bond is as facile as that of the aryl-aryl bond in 2-flourobiphenyl. Other versatile and unique properties of the studied compounds include a broad photoabsorption range (from 250 up to 1100 nm), a reduced HOMO-LUMO gap (1.26–1.68 eV), and a small singlet-triplet energy gap (3.65–5.68 kcal/mol).

One of the most impressive molecular dynamics among simple alkenes is the restricted rotation of C = C bonds at room temperature. In the rotational process, the transition state ($TS_{rot}$) is an open-shell singlet diradical with a 90° twisted conformation, and the geometry is virtually identical to that of the lowest-lying triplet state ($T_1$)[1]. Therefore, the rotational barrier and singlet-triplet gap ($\Delta E_{ST}$) are comparable, and those for ethene have been experimentally determined to be 65.9 ± 0.9 kcal/mol[2] and 58 ± 3 kcal/mol[3], respectively. The key to decreasing the rotational barrier lies in reducing the bond order of the C = C bond[4], and this strategy has often been applied in rotary molecular motors for rate acceleration[5]. In a push-pull alkene, the resonance effect allows the concerned C = C bond to exhibit partial C–C bond character[6]. For example, dimethyl 2-(*N,N*-dimethylamino)-2-ethylidenemalonate has a low rotational barrier ($\Delta G^{\ddagger}_{rot}$) of

approximately 8.7 kcal/mol (Fig. 1)[7]. Chichibabin's hydrocarbon[8], which contains a planar backbone, an elongated C = C bond (1.45 Å)[9] and remarkable biradical character[10], exhibits low $\Delta E_{ST}$ (calcd. 8.1 kcal/mol)[11], and $\Delta G^{\ddagger}_{rot}$ (11.4 kcal/mol)[12]. Sterically overcrowded bianthraquinodimethanes adopt a folded structure with an untwisted C = C bond and a regular bond length[13,14], yielding $\Delta G^{\ddagger}_{rot}$s exceeding 28 kcal/mol. Several highly twisted alkenes including bi[1,3-bis(dicyanomethylene)indan-2-ylidene][15], perchloro-9,9'-bifluorenylidene[16], 13,13'-bis(dibenzo[*a,i*]fluorenylidene) **1a**[17] ($\Delta E_{ST}$ = 3–4 kcal/mol[18], or 9.6 kcal/mol[19], tetra-*n*-octoxy-substituted bianthrone (TOBA, $\Delta E_{ST}$ = 5.7 kcal/mol)[20], and tetrafluorenofulvalene (TFF, $\Delta E_{ST}$ = 1.5 or 3.1 kcal/mol)[21] have been investigated but only the last three alkenes are electron spin resonance (ESR) active. The small $\Delta E_{ST}$ values for twisted TOBA[20] and TFF[21] are associated with the open-shell ground states whereas that of

[1]Department of Chemistry, National Cheng Kung University, 70101 Tainan, Taiwan. [2]Institute of Chemistry, Academia Sinica, 11529 Taipei, Taiwan. [3]Department of Chemistry, National Taiwan University, 10617 Taipei, Taiwan. [4]Department of Electrical Engineering, National Tsing Hua University, 30013 Hsinchu, Taiwan. [5]Department of Medical and Applied Chemistry, Kaohsiung Medical University, 80708 Kaohsiung, Taiwan. ✉e-mail: chop@ntu.edu.tw; mhchiang@chem.sinica.edu.tw; ytwuchem@mail.ncku.edu.tw

**1a** is related to the high-strain ground state $S_0$[22,23] and the low-lying $T_1$ state. The latter is stabilized by electron delocalization through the π-system[19,24] in addition to hyperconjugation[25].

In light of the information mentioned above, the chemistry, structures, molecular dynamics, and properties of a new series of highly strained 13,13'-bis(dibenzo[*a,i*]fluorenylidenes) **1a**–**1d** (see Fig. 1) and backbone expanded derivatives **2** and **3** are comprehensively explored herein to (1) improve or simplify the synthetic approach that would otherwise be too complicated and inefficient, (2) verify the roles of the π-extended substituents and backbones on the structure-property relationships, and (3) gain in-depth insight into $\Delta G^{\ddagger}_{rot}$ (for **1d** and **2**) and $\Delta E_{ST}$ using variable-temperature (VT) $^1$H NMR spectroscopy and a superconducting quantum interference device (SQUID), respectively. Methoxy and 4-tolylethynyl substituents in **1b**–**1d** are located at positions C-5 and C-8 of a DBF fragment (DBF = dibenzo[*a,i*]fluorene), which both significantly contribute to the frontier orbitals, $TS_{rot}$, and the $T_1$ state. All studied compounds lie in a core 9,9'-bifluorenylidene (9,9'-BF) structure, whose derivatives are promising candidates for use as electron acceptors in organic optoelectronics[26–28]. The studied 9,9'-BF based compounds were

strategically designed and should exhibit interesting electrochemical properties (vide infra). Although the synthesis and some properties of **3** have been reported[29], the structure was confirmed for the first time in this study using X-ray crystallography. Importantly, we found that the configurations of two semi-rigid helical substructures significantly affect the structural parameters and properties, which are discussed in the following sections.

## Results and discussion

### Preparations

The first synthesis of **1a** was documented about a century ago using straightforward protocols, such as the reaction of 13,13-dichloro-DBF with copper or heating di(1-naphthyl)carbinol in phosphoric acid[30]. An alternative approach involves the silver acetate-mediated oxidation of bis(13-dibenzo[*a,i*]fluorenyl) **7a** (61% yield)[18,19], which can be generated by thermolysis of 13-[di(1-naphthyl)methyl]-DBF in naphthalene (42%) or radical-mediated dimerization of 13*H*-DBF **4a** (32%)[31]. Bis(17-tetrabenzo[*a,c,g,i*]fluorenyl) **9** can be effectively furnished by the titanium-mediated dimerization of pre-isolated 17-litho-TBF (TBF = tetrabenzo[*a,c,g,i*]fluorene)[32]. Therefore, a one-pot procedure

**Fig. 1 | Selected examples of alkenes. a** A push-pull alkene. **b** Chichibabin's Hydrocarbon and derivatives. **c** Highly twisted alkenes. The C = C bond lengths exceeding 1.40 Å are indicated in red, while those falling below 1.40 Å are displayed in blue.

**Table 1 | Synthesis of alkenes 1, 2 and 3[a]**

| Substituent | Starting material | Product (yield, %) | Product (yield, %) |
|---|---|---|---|
| $R^1 = R^2 = R^3 = R^4 = H$ | **4a** | **7a** (90) | **1a** (78) |
| $R^1 = R^4 = 4$-TE, $R^2 = R^3 = H$ | **4b** | **7b** (80) | **1b** (35) |
| $R^1 = R^4 = $ OMe, $R^2 = R^3 = H$ | **4c** | **7c** (82) | **1c** (60) |
| $R^1 = $ OMe, $R^2 = R^3 = H$, $R^4 = 4$-TE | **4d** | **7d** (78) | **1d** (43) |
| $R^1$–$R^2 = C_4H_4$, $R^3 = R^4 = H$ | **5** | **8** (70) | **2** (72) |
| $R^1$–$R^2 = R^3$–$R^4 = C_4H_4$ | **6** or **10** | **9**(>61) | **3** (24) |

[a]*DMF = N,N-dimethylformamide, DMSO dimethyl sulfoxide, 4-TE 4-tolylethynyl.*

was developed herein where **7a** was directly synthesized from **4a** with a yield of 90% (Table 1). Aryl signals of **7a** in the [1]H NMR spectrum were broad and featureless at room temperature, presumably due to the rocking of the two DBF moieties around the central C–C single bond (Supplementary Fig. 1). Oxidative dehydrogenation of **7a** with a mixture of potassium *tert*-butoxide (^tBuOK) and *N,N*-dimethylformamide (DMF), which acted together as a radical generator[33], at 80 °C for 4 h yielded desired product **1a** (78%) as a dark green powder. Similarly, **1b**, **1c**, **1d**, **2** and **3** were prepared from **4b**, **4c**, **4d**, 15*H*-tribenzo[*a,c,i*] fluorene **5** and 17*H*-TBF **6** (or 8b*H*-TBF **10**), respectively (see Supplementary Figs. 2 and 3). The critical step was ^tBuOK-mediated oxidative dehydrogenation, and the reaction conditions were substrate dependent. Notably, treatment of **7a** and **9** with *n*-butyllithium followed by copper(II) chloride[34] did not convert them to **1a** and **3**, respectively. To maximize the chemical yields of **1b** (35%) and **1d** (43%), oxidative dehydrogenations of **7b** and **7d** were conducted in dimethyl sulfoxide (DMSO)[35] at room temperature for 1 h. Nonetheless, undesired greenish compounds were obtained and confirmed to be the dimeric derivatives of **1b** and **1d** based on their mass spectra. However, their structures cannot be identified using [1]H NMR spectroscopy due to broad and weak resonances. The conversion of methoxy-substituted compound **7c** to **1c** was very slow, and some of the starting material (approximately 20–30%) remained even after continuing the reaction for 16 h at 80 °C. For the synthesis of **2**, the oxidative dehydrogenation of **8** carried out in DMSO (72%) was more effective than that performed in DMF (30%). The highly stable (di)anion species of **9** is most likely responsible for the inefficient formation of **3** (24% yield), and most of the starting material (70%) was recovered[36]. Except for **3**, the aerobic solutions of the studied compounds were stable at room temperature for more than one week.

**Structural analysis**

The structures of **1a**, **1b**, **1c**, and **3** were verified using single-crystal X-ray crystallography (Supplementary Table 1). The most notable structural parameters were the twist angle ($\theta_1$) and length ($d$) of the central C = C bond, and the first term was verified by the interplanar angle of the two five-membered rings C and C′ (Fig. 2a). The planarization of the DBF moiety was determined by gauging the interplanar angle ($\theta_2$) of the two peripheral rings A and A′. Crystals of **1a** were composed of two independent molecules that had nearly identical structural parameters ($\theta_1 =$ approximately 50° and $d = 1.40$ Å, see Fig. 2b) and slightly bent DBF moieties ($\theta_2 = 10–16°$). Substituted derivatives **1b** [$\theta_1 = 58°$, $\theta_2 = 1°$

and 5°, $d = 1.431(4)$ Å] and **1c** [$\theta_1 = 55°$, $\theta_2 = 9°$ and 15°, $d = 1.404(3)$ Å] are more twisted than **1a** but an elongated C = C bond and flat DBF moieties were exclusively observed in the first compound. Compound **3** with a (*M,M*)-configuration (and its enantiomer) was identified, and the interplanar angle of rings D and D′ (defined as $\theta_3$) in the helical moiety was 53°, which was accompanied by a highly twisted DBF fragment ($\theta_2 = 39°$). The C = C bond in (*M,M*)-**3** was less distorted ($\theta_1 = 36°$) and slightly shorter [$d = 1.395(3)$ Å] relative to those of **1a**. Intermolecular interactions of these compounds are shown in Supplementary Figs. 4–7. The highly twisted structures of **1b** and **1c** are most likely associated with the molecular stacking effect because both half fragments of the molecule have intermolecular interactions with neighboring molecules. Although several short intermolecular carbon•••carbon contacts (≤3.40 Å) were observed in the molecular stacking of **1b** (Fig. 2c), they should not be the cause of the radical-radical interaction due to the long contact distance (vs. 3.14 Å in a reported example[37]) and the inconsistency with the spin-density plot (see below).

**Computational analysis**

Density functional theory [DFT[38], (U)B3LYP-D3/6-311 G + + **//6-31 G**] was used to examine the electronic structures, molecular geometries, and rotational dynamics of the studied compounds. Except for **1b**, the structures of all compounds obtained by X-ray crystallography were highly consistent with those of the corresponding $S_0$ state (see Supplementary Table 2). The slight inconsistency between the structural parameters for **1b** that were obtained experimentally [$d = 1.431(4)$ Å, $\theta_1 = 58°$, $\theta_2 = 1°$ and 5°] and computationally ($d = 1.413$ Å, $\theta_1 = 52°$, $\theta_2 = 14°$) is most likely due to perturbations in the intermolecular interactions in the former case, as previously mentioned. Although some functionals suggested that **1b** has an open-shell singlet (OS) biradical ground state (Supplementary Tables 3–5), the calculated $\Delta E_{ST}$ and/or structural parameters $d$ and $\theta_1$ strongly disagreed with those observed experimentally. In addition, the bond length $d$ in **1b** [exptl. 1.431(4) Å and calcd. 1.413 Å] is remarkably shorter than that in 6,6′-biindeno[1,2-*b*]anthracene [$d = 1.58(2)$ Å and 1.495 Å], a biaryl with high diradical character[39]. *syn*-**1d** and *anti*-**1d** with nearly identical structural parameters ($d = 1.409$ Å, $\theta_1 = 51°$, $\theta_2 = 14°$) are equally stable. $S_0$-**2** should contain four conformers when the *syn*- and *anti*-conformations as well as the slightly twisted fjord regions are taken into account. These conformers with ordinary structural parameters ($\theta_1 = 46°$, 50°; $d = 1.402–1.404$ Å) must coexist at room temperature due to the

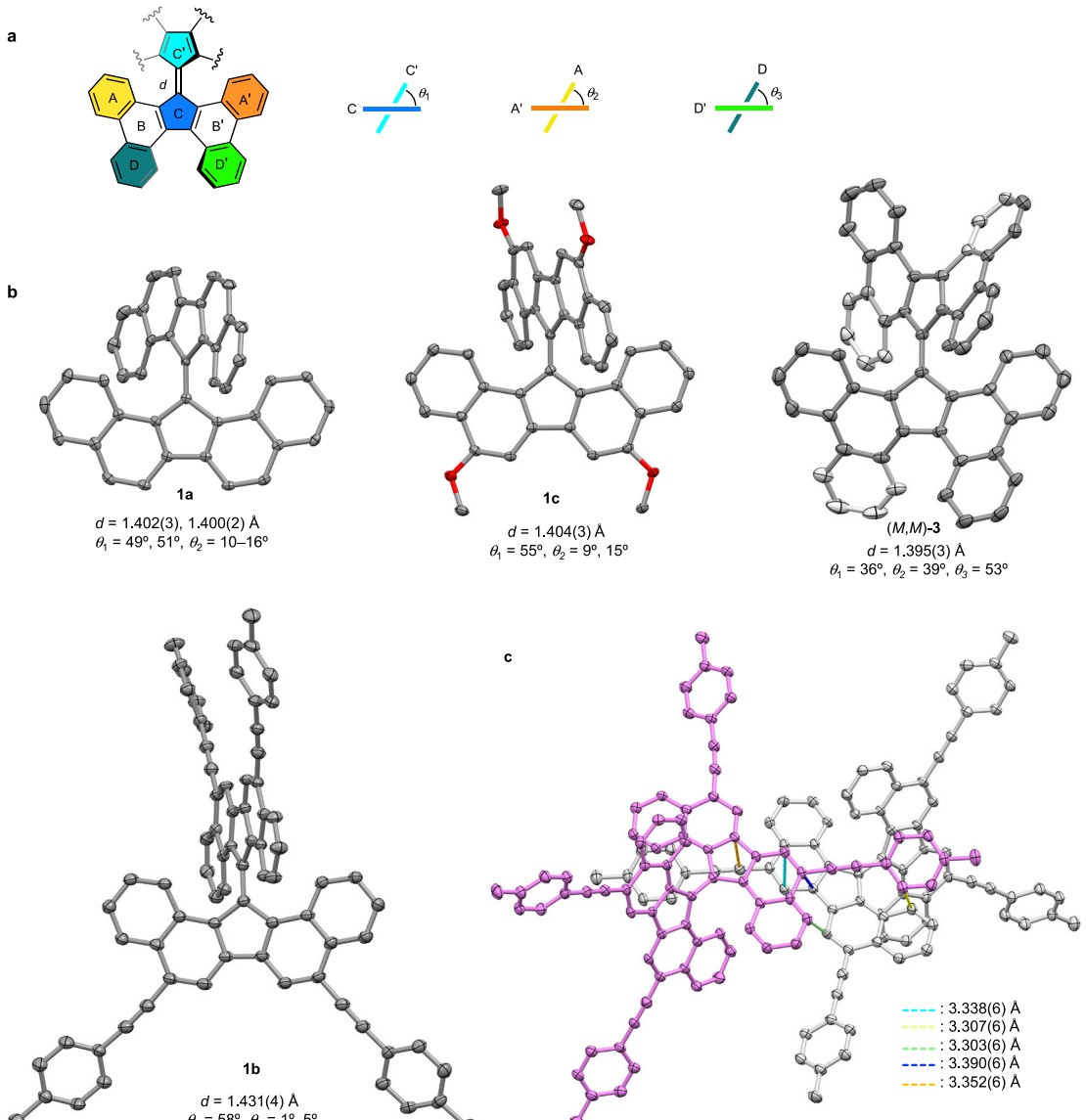

**Fig. 2 | Crystallographic structures. a** Definitions of structural parameters. **b** ORTEP plots of **1a**, **1b**, **1c** and **3** with 50% thermal ellipsoids. **c** Molecular stacking of **1b**. Hydrogen atoms are omitted for clarity.

barrierless inversion of the fjord region, low $\Delta H^{\ddagger}_{rot}$ (3.46 kcal/mol, Table 2), and the tiny energy difference (0.36 kcal/mol) between the most and least stable conformers (Supplementary Fig. 8). (*M,M*)-**3** is the representative diastereomer in this study, and the contributions of (*M,P*)-**3** ($\Delta H^{\ddagger}_{rot}$ = 2.65 kcal/mol, $\Delta H_{ST}$ = 3.51 kcal/mol) should be neglected (Supplementary Fig. 9).

Except for $T_1$-(*M,M*)-**3** ($\theta_1$ = 74°, $\theta_2$ = 22°), the $T_1$ state of the studied compounds exhibits a perpendicular conformation ($\theta_1$ = 86° or 90°) with an elongated bond $d$ (1.462–1.466 Å) and (nearly) planar DBF moieties ($\theta_2$ = 0°; 8° for $T_1$-**2**). The $\Delta H_{ST}$s obtained by DFT calculations are as folows: (*M,M*)-**3** (5.79 kcal/mol, see Table 2) > **1a** (5.52 kcal/mol) > **1c** (5.41 kcal/mol) > **1d** (4.26 kcal/mol) ≈ **2** (4.33 kcal/mol) > **1b** (3.19 kcal/mol). The low $\Delta H_{ST}$s for **1b** and **1d** most likely originate from additional stabilization of π-extended alkynyl moieties on $T_1$-**1b** and $T_1$-**1d**, respectively, as shown in their spin density plots (see below and Supplementary Fig. 10). In most cases, the molecular geometries of the TS$_{rot}$ and $T_1$ states are (nearly) identical (see Supplementary Table 2). Structural parameters of (*M,M*)-**3**-TS$_{rot}$ ($\theta_1$ = 82°, $\theta_2$ = 19°, $\theta_3$ = 41°, $d$ = 1.462 Å) and $T_1$-(*M,M*)-**3** ($\theta_1$ = 74°, $\theta_2$ = 22°, $\theta_3$ = 42°, $d$ = 1.462 Å) are slightly different, and their DBF moieties are highly twisted. The $\Delta H^{\ddagger}_{rot}$s

of the studied compounds were determined to be 2.35–5.13 kcal/mol (Table 2), indicating that the rotation of the C = C bond can be as facile as that of an aryl-aryl bond in a simple biphenyl, such as 2-flourobiphenyl ($\Delta G^{\ddagger}_{rot}$ 4.4 kcal/mol[40]).

**Photophysical properties**

The solution of **1a** exhibited a green color, which is associated with an absorption transition range from 260 to 866 nm (Table 2 and Fig. 3a) and a broad band centered at approximately 627 nm ($\lambda_{abs}$). $\lambda_{abs}$ of **1b**, **1c**, **1d**, **2** and **3** were located at 745 nm, 650 nm, 699 nm, 652 nm, and 642 nm, respectively, indicating that the alkynyl-substituted compounds exhibited remarkable bathochromic effects. The contributions of the alkynyl moieties to the frontier orbitals are shown in Supplementary Fig. 12. The estimated optical HOMO-LUMO gaps ($E_g^P$) were as follows: **1b** (1.40 eV) < **1d** (1.48 eV) < **1c** (1.56 eV) < **2** (1.60 eV) < **3** (1.66 eV) < **1a** (1.68 eV).

Unlike the 13-(9-anthryl)-13*H*-DBF radical[41] (ADBF radical in Fig. 3b, ε~100 at 1220 nm), no such signal was observed in the long-wavelength region (1200–1600 nm) for the studied compounds in the steady-state absorption spectra even when the measurements

**Table 2 | Selected properties of the studied compounds[a]**

|  | 1a | 1b | 1c | 1d | 2 | (M,M)-3 |
|---|---|---|---|---|---|---|
| $\lambda_{abs}$ (nm) | 627 | 745 | 650 | 699 | 652 | 642 |
| $E_g^P$ (eV) | 1.68 | 1.40 | 1.56 | 1.48 | 1.60 | 1.66 |
| $E_{ox}^{\frac{1}{2}}$ (V) | 0.53[irr] | 0.45[qr], 0.61[qr] | 0.15, 0.26 | 0.33[b] | 0.56[irr] | 0.65[irr] |
| $E_{red}^{\frac{1}{2}}$ (V) | −1.06, −1.36 | −0.81, −1.02 | −1.31, −1.54 | −1.05, −1.26 | −0.93, −1.20 | −0.94, −1.23 |
| $E_g^E$ (eV) | 1.59 | 1.26 | 1.46 | 1.38 | 1.49 | 1.59 |
| $\Delta H_{rot}^{\bullet}$ (kcal/mol) | 4.64[c] | 2.35[c] | 4.53[c] | 3.40[c] | ~3.46[c] | 5.13[c] |
| $\Delta E_{ST}$ (kcal/mol) | 5.32 [5.52[c]] | 3.65 [3.19[c]] | 5.36 [5.41[c]] | — [4.26[c]] | 4.32 [4.33[c]] | 5.68 [5.79[c]] |

[a]The electrochemical and photophysical properties of studied compounds were measured in $CH_2Cl_2$. $E_{ox}^{\frac{1}{2}}$ and $E_{red}^{\frac{1}{2}}$ are half-wave potentials of the oxidative and reductive waves, respectively, with potentials vs. Fc/Fc⁺ couple. qr = quasi-reversible. For an irreversible wave (irr), the potential was determined by its onset value. $E_g^E$ and $E_g^P$ are the electrochemical and optical HOMO-LUMO gaps, respectively.
[b]A one-step two-electron oxidation process.
[c]Data obtained by DFT calculation.

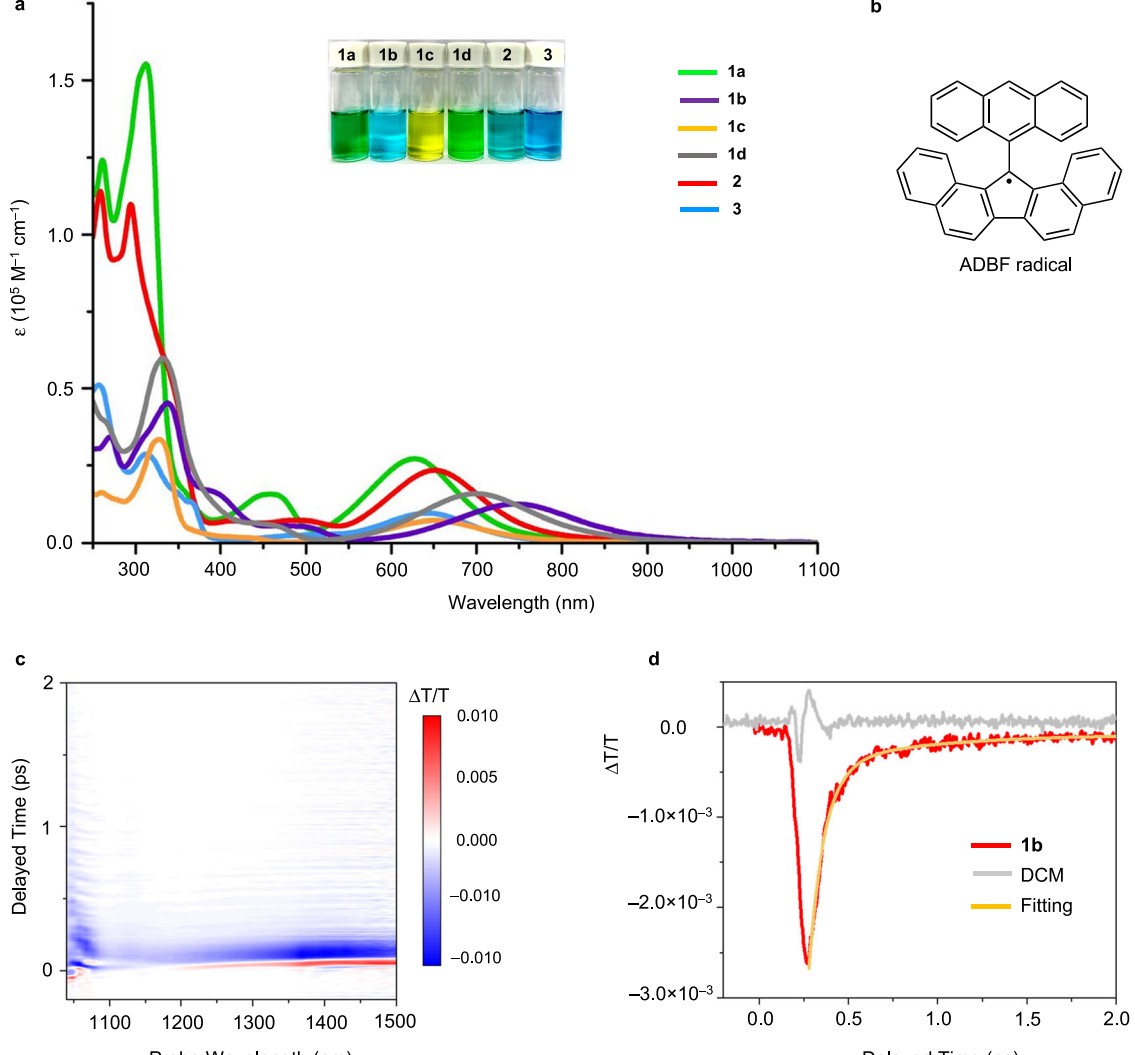

**Fig. 3 | Photophysical properties. a** Absorption spectra of studied compounds (10 µM in $CH_2Cl_2$). **b** Schematic structure of ADBF radical. **c** Visible-pump NIR-probe TA spectrum of **1b**. **d** Time trace of the TA spectra of **1b**. The time traces are obtained by averaging the ones at 1100–1200 nm. The lifetimes of 90 fs and 618 fs are acquired by fitting the bi-exponential function.

were performed with highly concentrated samples. The photo-dynamics were probed using transient absorption (TA) spectroscopy with visible pump and near-infrared (NIR) probe pulses (Supplementary Fig. 13). **1b** exclusively exhibited a significant excited-state absorption range from 1100–1500 nm (Fig. 3c). According to the characteristic absorption of the DBF radical[41], NIR electronic transitions are associated with diradical formation and can be correlated to the first singlet excited state ($S_1$) of **1b**. The time trace of the TA spectra indicated that the $S_1$ populations of **1b** decay with a lifetime of approximately 90 fs (Fig. 3d), revealing the ultrafast time scales of the

excited-state deactivation motions. A similar fluorescence decay time (ca. 90 fs) was also observed for 9,9′-BF[42], indicating that the excited-state relaxation is dominated by rotational dynamics that are not strongly affected by the substituents or the π-system of the backbone.

## Electrochemistry

The cyclic voltammogram of **1a** in $CH_2Cl_2$ exhibited one irreversible oxidation signal (0.53 eV) and two reversible reduction waves ($E^{1/2}_{red} = -1.06$ and $-1.36$ eV, see Table 2 and Supplementary Fig. 14). Similar to 9,9′-BFs[26,43], the addition of two electrons across the central C = C bond converted the studied compound into two (π-expanded) DBF anion units connected by a single bond. The driving forces in the reduction process should be steric strain relief and increased aromaticity to two $[4n + 2]$ π-electron systems. Two-electron (quasi) reversible oxidations were observed in the voltammograms of **1b**, **1c**, and **1d**. The 4-tolylethynyl substituents in **1b** and **1d** did act as π-extended electron-withdrawing groups to reduce both $E^{1/2}_{ox}$ and $E^{1/2}_{red}$. However, the electron-donating effects of the methoxy groups in **1c** decreased $E^{1/2}_{ox}$ but increased $E^{1/2}_{red}$. Compound **3** with its highly distorted molecular halves demonstrated the highest first oxidation potential. Consequently, the electrochemical HOMO-LUMO gaps ($E^E_g$) of the studied compounds are ranked in the following order: **1b** (1.26 eV) < **1d** (1.38 eV) < **1c** (1.46 eV) < **2** (1.49 eV) < **3** = **1a** (1.59 eV), in line with that of $E^P_g$.

## VT $^1$H NMR spectra

At room temperature, **1a** exhibits well-resolved resonances in the $^1$H and $^{13}$C NMR spectra, and thermal treatments progressively broadened the spectral lines[19] due to the small $\Delta E_{ST}$ value. Therefore, **1b** and **2** must have even smaller $\Delta E_{ST}$ values. All signals for **1b** (except 4-tolyl groups) as well as 6-H, 8-H, 10-H, and 13-H of **2** are highly broad in the $^1$H NMR spectra recorded at room temperature (Fig. 4a). The hydrogen atoms, which exhibited line broadening, are bound to the carbon atoms bearing remarkable spin density in the $T_1$ state (Fig. 4b). Some broad signals also appeared in the $^{13}$C NMR spectra of **1b**, **1d**, and **2**.

VT $^1$H NMR experiments using unsymmetrical compounds **1d** (183–303 K) and **2** (183–413 K) were performed (see Supplementary Information p. 43 and 45). Upon cooling, the signals exhibited progressive line sharpening without any significant decoalescence. The phenomenon is most likely due to the following three possibilities: (1) identical resonances of the *syn-* and *anti-*isomers, (2) the amount of energetically unstable isomer being too low to be detected, and (3) the C = C bond rotational rate at 183 K being faster than the timescale of $^1$H NMR spectroscopy. The first two options are unlikely, and the energy difference between the two isomers is less than 0.13 kcal/mol (see Supplementary Information p. 33), which results in an unstable isomer with a Boltzmann population greater than 40% at 183 K. In the VT $^1$H NMR spectra of **3**, peaks with progressive line broadening were observed at temperatures higher than 303 K.

## Magnetic properties

The temperature-dependent magnetic susceptibilities ($\chi_M$) for powdered samples of **1a**, **1b**, **1c**, **2**, and **3** were obtained using a SQUID magnetometer. The $\chi_M T$ curves for all compounds exhibited decreasing trends when the temperature was decreased to 2 K, suggesting a singlet ground state (Supplementary Fig. 15). The value of $\Delta E_{ST}$ was estimated by fitting the temperature-dependent $\chi_M T$ data using the Bleaney-Bowers equation[44]. Satisfactory fits were achieved, and **1b** exhibited the smallest $\Delta E_{ST}$ (3.65 kcal/mol) compared to the other compounds (4.32–5.68 kcal/mol, Table 2).

The ESR spectra of the powdered samples exhibited a featureless broad signal centered at g = 2.00 without $\Delta M$s = ±2 transitions at room temperature (Supplementary Fig. 16)[45]. This spectral feature can be rationalized by the weak spin–spin dipole interaction between the two radicals[46]. The signal intensity of all studied compounds increases as the

temperature increases from 300 to 400 K, and the most enhancement was observed for **1b**. In contrast to other compounds, **1b** exhibited the strongest intensity at room temperature, indicating the smallest $\Delta E_{ST}$, the greatest spin delocalization over the molecule, including the 4-tolylethynyl substituents, and possibly a second-order contribution from dipolar interactions. A half-field signal attributed to the forbidden transition ($\Delta M_S = \pm 2$) of the triplet state of **1b** in the 2-methyltetrahydrofuran glass phase was observed at a temperature of 7 K, confirming the triplet state is slightly higher in energy (Fig. 4c). The ESR spectrum of **1b** in the toluene glass state at 150 K displayed a hyperfine structure (Fig. 4d). A satisfactory fit to the experimental data yielded an isotropic signal at g = 2.0029 as well as a signal at g = 2.003 with $A_H = -13.9948$ and $-18.5058$ MHz. The hyperfine coupling is attributed to the interaction of the radical with the 6-H/7-H atoms based on the spin density plot of **1b**. Given the easily accessible triplet state, the data was also simulated using the two unpaired electron system including the anisotropic terms, yielding a satisfactory fit with g = 2.003 with D = 25 and E = 6.8 MHz. The hyperfine splitting due to radical/hydrogen or radical/radical interactions (or both together) cannot be completely confirmed as the dipolar and nuclear hyperfine terms have a similar order of magnitude. The ESR spectrum of the solution sample of **1b** displays the characteristics of a monoradical doublet state ( >150 K, Fig. 4e). Similar to nitroxide[11,47] and dithiazole diradicals[48,49], **1b** also possesses strong intramolecular magnetic exchange coupling ($-2J$) that is much greater than the hyperfine parameter ($|-2J| \gg |A|$), creating the monoradical-like hyperfine feature. A broad signal is dominant due to the fast molecular tumbling effect as the temperature increase to more than 150 K. The observed hyperfine structure of **1b** was obscured in the solid-state spectra, presumably due to the broadening caused by intermolecular dipolar coupling.

In summary, the syntheses, X-ray crystallographic structures, and properties of several highly twisted alkenes were systematically investigated. The target molecules were prepared from the corresponding benzofluorene derivative through titanium-mediated dimerization and subsequent oxidative dehydrogenation. The properties of the studied compounds are strongly associated with the substituents, the π system, and the molecular geometry. Compound **1b** possessed the smallest $\Delta H^{\ddagger}_{rot}$ (2.35 kcal/mol), $\Delta E_{ST}$ (3.65 kcal/mol), and $E^E_g$ (1.26 eV) values, which were associated with the stabilization effect of the 4-tolylethynyl substituents. Such a low $\Delta H^{\ddagger}_{rot}$ value indicates that the rotation of the C = C bond could be as facile as an aryl-aryl bond in a simple biphenyl. The molecular geometry of **1b** has three essential structural features, namely a highly twisted and elongated C = C bond ($\theta_1 = 58°$, d = 1.43 Å), an extended π-system, and a quasi-planar molecular half-fragment ($\theta_2 < 5°$). The final is very close to those in the $TS_{rot}$ and the $T_1$ state ($\theta_2 = 0°$) and important to buttress the highly twisted C = C bond. The π-extended substituent 4-tolylethynyl group (in **1b**) is superior to the π-expanded backbone (in **2** and **3**) not only in reducing the $E_g$ height but also for stabilizing $TS_{rot}$ and $T_1$. Exclusively, **1b** exhibits significant excited-state absorption in the TA spectrum, denoting the formation of the stabilized $S_1$ state with a diradical character. With this successful synthetic approach and essential molecular design information, applications of functionalized π-extended twisted alkenes as organic materials and molecular machines are currently being explored. Despite the progress made in increasing the rotational rates of these alkenes, constructing a molecular motor with unidirectional rotation remains a significant challenge[5].

## Methods

### Synthesis

**General procedure for synthesis of compounds 7a–d, 8, and 9.** To a solution of a fluorene derivative (1.00 mmol) in anhydrous THF (10 mL) at −78 °C, a solution of $^n$BuLi (1.10 mmol) was dropwise added. After being stirred at the same temperature for 1 h, the solution was treated with a solution of $TiCl_4$ (1.00 mmol) in toluene. The reaction mixture

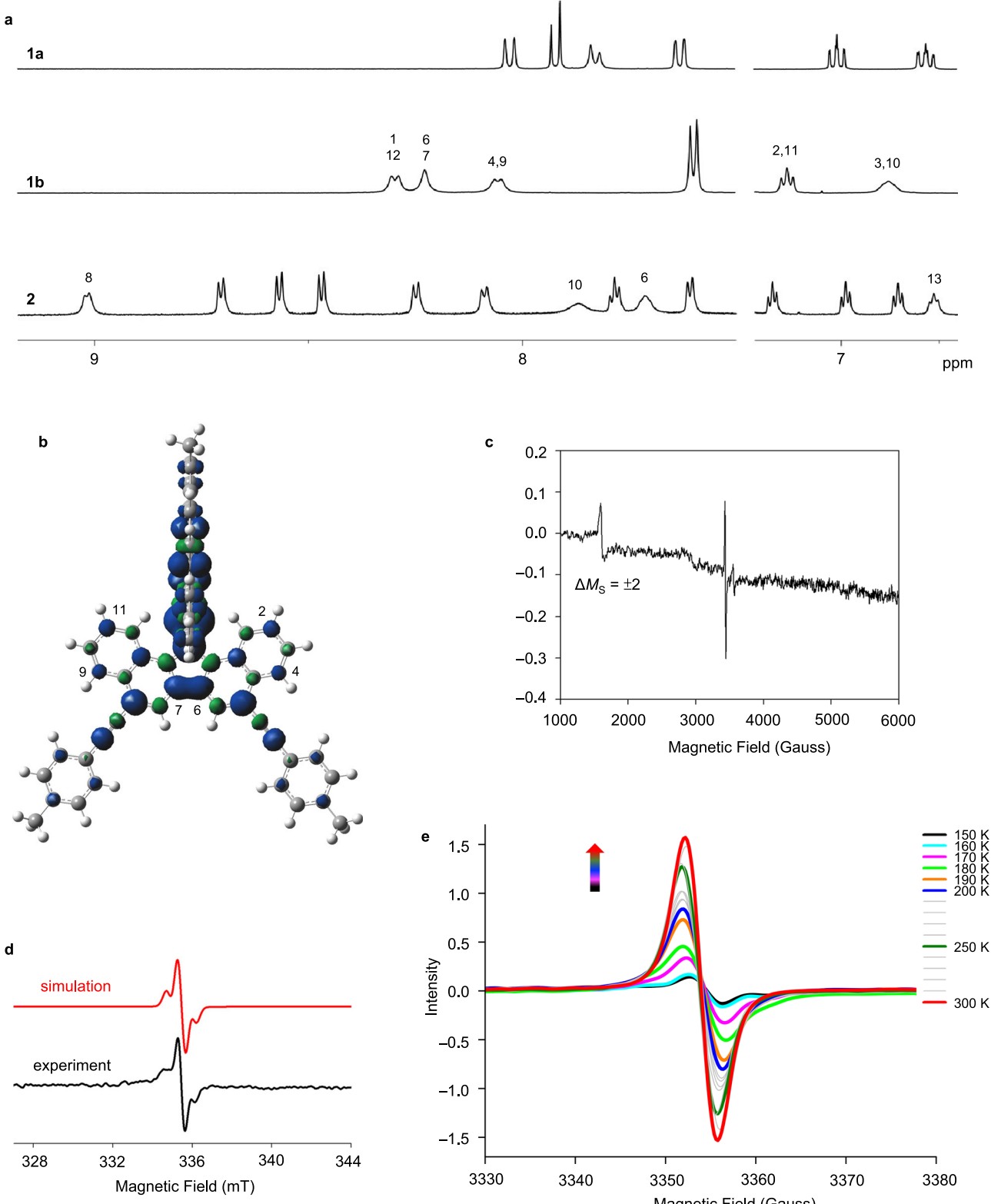

**Fig. 4 | Magnetic properties. a** $^1$H NMR spectra (selected regions) of **1a, 1b**, and **2** in CDCl$_3$ at room temperature. **b** Spin-density plot of $T_1$-**1b**. **c** $\Delta M_S = \pm 2$ forbidden transition of **1b** in 2-MeTHF at 7 K. **d** ESR spectrum of **1b** at 150 K in toluene and the simulated spectrum. **e** VT ESR spectra of **1b** in a toluene solution.

was slowly warmed to room temperature and stirred overnight. The product was collected by filtration, rinsed with water (20 mL) and dried in vacuo. In some cases, additional products can be obtained from the filtrate by chromatography on silica gel. For details, see Supplementary Information.

**Procedure for synthesis of highly twisted alkenes 1a–d, 2, and 3.** Compounds **1a–d**, **2**, and **3** were prepared from **7a–7d**, **8** and **9**, respectively, by the $^t$BuOK-mediated oxidative dehydrogenation. The reaction conditions were substrate dependent. For details, see Supplementary Information.

**Characterization.** NMR spectra were recorded on Brucker 400 MHz ($^1$H, 400 MHz), 500 MHz ($^1$H, 500 MHz; $^{13}$C, 125 MHz), and 700 MHz ($^1$H, 700 MHz; $^{13}$C, 175 MHz) spectrometers. Chemical shifts are reported in δ (ppm) relative to the solvent peak: chloroform-$d$ (CDCl$_3$: $^1$H, 7.26 ppm; $^{13}$C, 77.00 ppm), dichloromethane-$d_2$ (CD$_2$Cl$_2$: $^1$H, 5.32 ppm; $^{13}$C, 53.80 ppm) and 1,1,2,2-tetrachloroethane-$d_2$ (C$_2$D$_2$Cl$_4$: $^1$H, 6.00 ppm; $^{13}$C, 73.78 ppm). High-resolution mass spectra (HRMS) were obtained on JEOL JMS-700, JEOL AccuTOF GCx-plus or Bruker New ultra-fleXtreme Mass Spectrometers. Melting points were determined with a Büchi melting point apparatus B545 and are uncorrected.

**X-Ray crystallography.** Crystals of **1a**, **1b** and **1c** were obtained by slow diffusion of methanol in the dichloromethane solutions at room temperature. Crystals of **3** were grown from dichloromethane by slow evaporation of the solvent at room temperature. Single-crystal X-ray diffraction was performed on a Bruker D8 Venture Single-Crystal X-Ray Diffractometer or Rigaku XtaLAB Synergy DW, and the data were collected and processed by using a PHOTON III detector and a HyPix-Arc 150° curved hybrid photon counting X-ray detector, respectively.

**DFT calculations.** All of the DFT calculations were calculated using the B3LYP functional[50] with the addition of D3 dispersion correction[51] unless stated otherwise. The double-z quality 6-31 G** basis set[52] was chosen for geometry optimization and thermodynamic correction (ZPE, $H_{vib}$, $H_{trans}$, $H_{rot}$), while the triple-z quality 6-311 + + G** basis set was used for obtaining more accurate electronic energies ($E_{elec}$). The reported energies are enthalpies, which were calculated based on the following equation:

$$H = E_{elec} + ZPE + (H_{vib} + H_{trans} + H_{rot} + RT)$$

**Cyclic voltammetry.** The electrochemical measurements were carried out with a CH Instruments 621E potentiostat at room temperature, using a three-electrode cell under a nitrogen atmosphere. Cyclic voltammograms (CV) were recorded with a glassy carbon working electrode, a Pt counter electrode and a Ag/AgNO$_3$ (0.1 M AgNO$_3$ and 0.1 M N($^n$Bu)$_4$ClO$_4$ in acetonitrile) reference electrode in an electrolyte solution of 0.1 M TBAPF$_6$ (tetrabutylammonium hexafluorophosphate) in CH$_2$Cl$_2$ with a scan rate of 0.1 V/s. The potential was externally calibrated against the ferrocene/ferrocenium couple.

**Photophysical Properties.** Steady-state absorption spectra were measured by Perkin Elmer (Lambda 950) spectrophotometer. The detailed setups of the light source and transient absorption spectroscopy are described in Supplementary Information.

**Magnetic properties.** The temperature-dependent molar magnetic susceptibility of crystalline solid samples was measured on a Quantum Design SQUID-VSM magnetometer with an applied field of 5000 G or 2000 G. The magnetic susceptibility data were corrected for the diamagnetic contribution of the capsule and the sample holder. Molar susceptibility data were corrected for diamagnetic contribution of Pascal's constants[53], The ESR measurements were obtained with a Bruker EMX-10/12 and EPR-plus X-band (9.4 GHz) digital EPR spectrometer with Bruker N$_2$-temperature controller. Spectral analysis and simulations were performed using the EasySpin program.

## Data availability
All experimental procedures, characterization data and NMR spectra are available within the article and the Supplementary Information. The X-ray crystallographic data for structures reported in this study have been deposited at the Cambridge Crystallographic Data Centre (CCDC), under deposition numbers 2116430 (**1a**), 2204920 (**1b**), 2204921 (**1c**), and 2116431 (**3**). These data can be obtained free of charge from The Cambridge Crystallographic Data Centre via www.ccdc.cam.ac.uk/data_request/cif.". Supplementary Data 1 (the Cartesian coordinates of optimized structures) are provided with this paper.

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

## Acknowledgements

This work was financially supported by the National Science and Technology Council of Taiwan (MOST-110-2113-M-006-012-MY3, Y.T.W.; MOST-110-2639-M-002-001-ASP, P.T.C. and MOST-111-2113-M-006-010-MY3, M.J.C.) and Academia Sinica (AS-iMATE-111-21, M.H.C.). Instrumentation Centers at the National Tsing Hua University and the National Cheng Kung University are acknowledged for the X-ray structure analyses.

## Author contributions

Y.T.W. conceived the idea and designed the experiments. H.W.K. undertook the synthesis and analysis of all compounds, as well as the measurement of photoabsorption and electrochemical properties. M.H.C. directed investigations of magnetic properties. Y.C.L. conducted the ESR and SQUID measurements. M.J.C. provided guidance for the computational studies, while C.T.H. and W.K.S. executed the DFT calculations. P.T.C. and S.D.Y. supervised photodynamic investigations. B.H.C., C.H.L. and Y.C.W. measured the TA spectra and analyzed data. M.H.C, P.T.C. and Y.T.W. co-wrote the manuscript. All authors discussed the results and commented on the manuscript.

## Competing interests

The authors declare no competing interests.
