## [Peer Review File · Nature Communications]

Synthesis, Structural Analysis, and Properties of Highly Twisted Alkenes 13,13'-Bis(dibenzo[a,i]fluorenylidene) and Its DerivativesReviewers' Comments:

Reviewer #1:

Remarks to the Author:

This manuscript describes highly twisted 13,13'-bis(dibenzo[a,i]fluorenylidene) derivatives. The results are interesting and the manuscript is well written, although the organization of the manuscript could be improved. The study is scientifically sound and the characterization of the compounds is exhaustive. However, this work lacks the novelty aspect that is needed to be published in Nature Comm. In fact, the concept developed in this manuscript has been exploited several years ago (see J. Am. Chem. Soc. 2014, 136, 36, 12784–12793, J. Am. Chem. Soc. 2012, 134, 35, 14513–14525, J. Mol. Model 2012, 18, 5089–5095, ACIE 2016, 55, 14600–14605). The compounds presented here are interesting and novel, but the concept novelty is lacking. Thus, I suggest the authors to submit their manuscript to another journal.

Reviewer #2:

Remarks to the Author:

The manuscript from Wu and colleagues describes syntheses of pi-extended 9,9'-bifluorenylidenes (BFs), 1-3. The parent system 1a has been known for nearly a century, but despite its unique properties, it has received relatively limited attention, and few of its derivatives are known. The authors develop a practical synthesis 1a, its substituted derivatives, and two pi-extended systems, 2 and the previously reported 3. The work is comprehensive and includes crystal structure analyses, spectroscopy (including TA spectroscopy, which is relatively rarely explored for open-shell molecules), and a significant amount of computational work, much of which is hidden in the Supplementary Information. The latter is of exemplary quality and is uniquely extensive (175 pages).

I like the paper very much not only because it addresses highly interesting chemistry but also because it is really well written, providing a lot of detail in a very succinct form. The reference coverage is very good, going back to Magidson's classic paper and describing modern advances. A related paper, which provides complementary insight into the relationship between bond strength and open-shell character, is currently available as a preprint at <https://doi.org/10.21203/rs.3.rs-2587889/v1> (accepted in Nature Chemistry). The authors may wish to cite this work in their revised manuscript.

To conclude, even though some of the π -extended BF motifs have been previously reported, the present work provides a very useful synthetic method that may be generally applicable to a variety of more elaborate targets, and provides new knowledge on the structure, electronic properties, and dynamics of twisted alkenes. I can therefore recommend publication of this manuscript in Nat. Comm. following minor revision.

Other comments:

"Upon cooling, the signals exhibited progressive line sharpening without any significant decoalescence, indicating that the syn- and anti-isomers were indistinguishable in the studied temperature range. This phenomenon may originate from the two isomers exhibiting identical signals and/or the C=C bond rotation rate at 183 K being faster than the timescale of ^1H NMR spectroscopy."

1. "sharpening"

2. The description could be improved. The isomers are not "indistinguishable", rather, their coexistence cannot be proven by the available data. Technically, there are three possibilities. Two have been mentioned above by the authors: (1) identical signals (unlikely), (2) rapid dynamics (most likely). One more possibility is that (3) just one conformer is populated because of a sufficiently large energy difference. Possibilities 2 and 3 could be verified by comparing NMR data with DFT predictions and estimating the expected populations/coalescence temperatures. Such an analysis does not need to be quantitative to be conclusive.

The use of empirical dispersion could be indicated in the manuscript (to reflect the content of the SI file).

“by the substitutes or the p-system of the backbone”
substituents

SI file:

Providing Cartesian coordinates as separate files (perhaps deposited externally), could be a more practical solution than compiling them in the SI document.

1D COSY spectra require a more detailed caption (p. SI-156, SI-160, etc.) It is not clear if these are cross sections of the 2D experiment or selCOSY experiments, what is the excited/selected signal, what is the significance of phases, and why were these experiments needed.

Reviewer #3:

Remarks to the Author:

The authors report an unprecedented synthesis and characterization of highly interesting twisted alkenes, introducing the concept that the rotation of a C=C bond is as facile as a C-C bond.

The manuscript is very well written, concise, with a very interesting hypothesis, well executed synthesis and characterization, being the conclusions solid and the results quite sound.

Thus, I would recommend publication, with a minor comment. I would appreciate a connection of the rotation of the bonds to the molecular machines.

Replies to reviewers' comments:

Reviewer #1 (Remarks to the Author):

General Comment: This manuscript describes highly twisted 13,13'-bis(dibenzo[a,i]-fluorenylidene) derivatives. The results are interesting and the manuscript is well written, although the organization of the manuscript could be improved. The study is scientifically sound and the characterization of the compounds is exhaustive. However, this work is lack the novelty aspect that is needed to be published in Nature Comm. In fact, the concept developed in this manuscript has been exploited several years ago (see J. Am. Chem. Soc. 2014, 136, 36, 12784–12793, J. Am. Chem. Soc. 2012, 134, 35, 14513–14525, J. Mol. Model 2012, 18, 5089–5095, ACIE 2016, 55, 14600-14605). The compounds presented here are interesting and novel, but the concept novelty is lacking. Thus, I suggest the authors to submit their manuscript to another journal.

Reply: We thank the reviewer for this comment. The important achievements of this work are as follows: 1) An improved/simplified synthetic approach to the title compounds has been developed; 2) The roles of the π -extended substituents and backbones on the molecular geometry and properties have been confirmed; 3) The rotation of the central C=C bond in the studied compounds is as facile as that of an aryl-aryl bond in 2-fluorobiphenyl; 4) The stabilized TS_{rot} of **1b** with diradical character was exclusively captured using transient absorption spectroscopy; 5) The configurations of the two semi-rigid helical substructures in **3** significantly affect the structural parameters and properties; and 6) The title compounds are promising candidates for use as electron acceptors in organic optoelectronics. Other versatile and unique properties of the studied compounds include a broad photoabsorption range (from 250 to 1100 nm), a reduced HOMO-LUMO gap (1.26–1.68 eV) as well as a small singlet–triplet energy gap (3.65–5.68 kcal/mol).

Reviewer #2 (Remarks to the Author):

General Comment:

The manuscript from Wu and colleagues describes syntheses of π -extended 9,9'-bifluorenylidenes (BFs), 1-3. The parent system 1a has been known for nearly a century, but despite its unique properties, it has received relatively limited attention, and few of its derivatives are known. The authors develop a practical synthesis 1a, its substituted derivatives, and two π -extended systems, 2 and the previously reported 3. The work is comprehensive and includes crystal structure analyses, spectroscopy (including TA spectroscopy, which is relatively rarely explored for open-shell molecules), and a significant amount of computational work, much of which is hidden in the Supplementary Information. The latter is of exemplary quality and is uniquely extensive (175 pages).

I like the paper very much not only because it addresses highly interesting chemistry but also because it is really well written, providing a lot of detail in a very succinct form. The reference coverage is very good, going back to Magidson's classic paper and describing modern advances. A related paper, which provides complementary insight into the relationship between bond strength and open-shell character, is currently available as a preprint at <https://doi.org/10.21203/rs.3.rs->

2587889/v1 (accepted in Nature Chemistry). The authors may wish to cite this work in their revised manuscript.

To conclude, even though some of the π -extended BF motifs have been previously reported, the present work provides a very useful synthetic method that may be generally applicable to a variety of more elaborate targets, and provides new knowledge on the structure, electronic properties, and dynamics of twisted alkenes. I can therefore recommend publication of this manuscript in Nat. Comm. following minor revision.

Reply:

1) We are grateful to the reviewer for the supportive comments and have made the necessary revisions to incorporate the suggestions.

2) The reference suggested by Reviewer #2 has been cited in the revised manuscript (Figure 1 and ref. 21). The paragraph in the Introduction has been rewritten as follows:

"Several highly twisted alkenes including bi[1,3-bis(dicyanomethylene)indan-2-ylidene]¹⁵, perchloro-9,9'-bifluorenylidene¹⁶, 13,13'-bis(dibenzo[*a,i*]fluorenylidene) **1a**¹⁷ ($\Delta E_{ST} = 3-4$ kcal/mol¹⁸ or 9.6 kcal/mol¹⁹), tetra-*n*-octoxy-substituted bianthrone (TOBA, $\Delta E_{ST} = 5.7$ kcal/mol)²⁰, and tetrafluorenofulvalene (TFF, $\Delta E_{ST} = 1.5$ or 3.1 kcal/mol)²¹ have been investigated but only the last three alkenes are electron spin resonance (ESR) active. The small ΔE_{ST} values for twisted TOBA²⁰ and TFF²¹ are associated with the open-shell ground states whereas that of **1a** is related to the high-strain ground state S_0 ^{22,23} and the low-lying T_1 state. The latter is stabilized by electron delocalization through the π -system^{19,24} in addition to hyperconjugation²⁵. "

Other comments:

Comment #1:

"Upon cooling, the signals exhibited progressive line sharpening without any significant decoalescence, indicating that the *syn*- and *anti*-isomers were indistinguishable in the studied temperature range. This phenomenon may originate from the two isomers exhibiting identical signals and/or the C=C bond rotation rate at 183 K being faster than the timescale of 1H NMR spectroscopy."

1. "sharpening"

2. The description could be improved. The isomers are not "indistinguishable", rather, their coexistence cannot be proven by the available data. Technically, there are three possibilities. Two have been mentioned above by the authors: (1) identical signals (unlikely), (2) rapid dynamics (most likely). One more possibility is that (3) just one conformer is populated because of a sufficiently large energy difference. Possibilities 2 and 3 could be verified by comparing NMR data with DFT predictions and estimating the expected populations/coalescence temperatures. Such an analysis does not need to be quantitative to be conclusive.

Reply:

1) The typo has been corrected.

2) The sentences have been rewritten as follows: The phenomenon is most likely due to the following three possibilities: 1) identical resonances of the *syn*- and *anti*-isomers, 2) the amount of energetically

unstable isomer being too low to be detected, and 3) the C=C bond rotational rate at 183 K being faster than the timescale of ^1H NMR spectroscopy. The first two options are unlikely, and the energy difference between the two isomers is less than 0.13 kcal/mol (see Supplementary Information p. 33), which results in an unstable isomer with a Boltzmann population greater than 40% at 183 K.

Comment #2:

The use of empirical dispersion could be indicated in the manuscript (to reflect the content of the SI file).

Reply: The functional has been corrected to B3LYP-D3/6-311++G**//6-31G**. This information has been added to the Methods section.

"All of the DFT calculations are calculated using the B3LYP functional⁵⁰ with the addition of D3 dispersion correction⁵¹ unless stated otherwise."

Comment #3:

"by the substitutes or the p-system of the backbone"
substituents

Reply: The typo has been corrected.

SI file:

Comment #4:

Providing Cartesian coordinates as separate files (perhaps deposited externally), could be a more practical solution than compiling them in the SI document.

Reply: The Cartesian coordinates have been removed from the Supplementary Information and deposited as the Supplementary Data.

Comment #5:

1D COSY spectra require a more detailed caption (p. SI-156, SI-160, etc.) It is not clear if these are cross sections of the 2D experiment or selCOSY experiments, what is the excited/selected signal, what is the significance of phases, and why were these experiments needed.

Reply: 2D COSY spectra provided sufficient information for signal assignment. Therefore, the selective 1D spectra have been deleted for clarity.

Reviewer #3 (Remarks to the Author):

General Comment: The authors report an unprecedented synthesis and characterization of highly interesting twisted alkenes, introducing the concept that the rotation of a C=C bond is as facile as a C-C bond. The manuscript is very well written, concise, with a very interesting hypothesis, well executed synthesis and characterization, being the conclusions solid and the results quite sound. Thus, I would recommend publication, with a minor comment. I would appreciate a connection of the rotation of the bonds to the molecular machines.

Reply: We are grateful to the reviewer for the supportive comments and have made the necessary revision to incorporate the suggestion. The final sentence in the conclusion has been changed as follows: "With this successful synthetic approach and essential molecular design information, applications of functionalized π -extended twisted alkenes as organic materials and molecular machines are currently being explored. Despite the progress made in increasing the rotational rates of these alkenes, constructing a molecular motor with unidirectional rotation remains a significant challenge⁵."